# The Characteristics of Self-Hydration and Carbonation Reaction of Coal Ash from Circulating Fluidized-Bed Boiler by Absorption of CO_2_

**DOI:** 10.3390/ma16155498

**Published:** 2023-08-07

**Authors:** Woong-Geol Lee, Myong-Shin Song, Seung-Min Kang

**Affiliations:** 1Research Center of Advanced Convergence Processing on Materials, Kangwon National University, Samcheok 25913, Republic of Korea; 2Technical Research Center, KC Green Materials Co., Ltd., Samcheok 25961, Republic of Korea

**Keywords:** CFBC coal ash, CO_2_ absorption, carbonation reaction, self-hydration

## Abstract

The by-products of the circulating fluidized-bed boiler combustion (CFBC) of coal exhibit self-hardening properties due to the calcium silicates generated by the reaction between SiO_2_ and CaO, and the ettringite generated by the reaction of gypsum and quicklime with activated alumina. These reactions exhibit tendencies similar to that of the hydration of ordinary Portland cement (OPC). In this study, the self-hydration and carbonation reaction mechanisms of CFBC by-products were analyzed. These CFBC by-products comprise a number of compounds, including Fe_2_O_3_, free CaO, and CaSO_4_, in large quantities. The hydration product calcium aluminate (and/or ferrite) of calcium aluminate ferrite and sulfate was confirmed through instrumental analysis. The CFBC by-products attain hardening properties because of the carbonation reaction between calcium aluminate ferrite and CO_2_. This can be identified as a self-hardening process because it does not require a supply of special ions from the outside. Through this study, it was confirmed that CFBC by-products generate CaCO_3_ through carbonation, thereby densifying the pores of the hardened body and contributing to the development of compressive strength.

## 1. Introduction

Circulating fluidized-bed boiler combustion (CFBC) has been in use in more than 10% of power plants since the mid-2000s [1,2,3,4,5]. When limestone (CaCO_3_) and coal fuel are subjected to CFBC at the same time, they are decomposed into CaO and CO_2_, whereas desulfurization gypsum is generated by the reaction between SOx and oxygen in the boiler [1,6,7]. Limestone is generally added in large quantities to improve the desulfurization efficiency, which causes the presence of unreacted CaO [8,9].

CFBC by-products have limitations in being used as construction materials, such as pulverized combustion (PC) fly ash (PCA). It is difficult to use them as substitutes for cement because of exothermic reactions involving the quicklime component and the expansion cracks caused by the hydration reaction of quicklime. Therefore, CFBC by-products are mostly landfilled. However, this workaround has resulted in a number of problems, such as economic losses due to the massive treatment cost and the saturation of landfills [10,11]. Although most of these by-products can be used as solidifying agents or cover materials, the consumption is insignificant compared to the amount generated.

According to previous studies, CFBC by-products exhibit self-hardening properties because of the calcium silicates generated by the reaction between SiO_2_ and CaO, and the ettringite generated by the reaction of gypsum and quicklime with activated alumina [12,13,14,15,16]. These by-products are typically composed of calcium silicate hydrates (C-S-H), ettringite, and Ca(OH)_2_, which are similar to the hydrates of ordinary Portland cement (OPC). In addition, there are research findings that show that CFBC by-products have the potential to be used as substitutes for cement because they exhibit highly pozzolanic reactions because of their high SiO_2_ and Al_2_O_3_ content [17]. Furthermore, research on the use of these by-products as non-sintered binders has been actively conducted [18].

In this study, the self-hydration and carbonation reaction mechanisms of circulating fluidized-bed boiler fly ash (CFBA) by absorption of CO_2_, in terms of the formation of ettringite (Ca_6_Al_2_(SO_4_)_3_(OH)_2_·26H_2_O) and carboferrite hydrates (3CaO·Fe_2_O_3_·CaCO_3_·12H_2_O) during the hydration reaction were investigated using the component characteristics of Fe_2_O_3_, free CaO, and CaSO_4_.

## 2. Experiment

### 2.1. Materials

In this study, the coal combustion by-products from CFBC in the Samcheok Green Power of Korea Southern Power Co., Ltd., Gangwon, Republic of Korea, i.e., CFBA and circulating fluidized-bed boiler bottom ash (CFBBA), were used. The physicochemical properties and crystalline structures of the two materials are shown in Table 1 and Figure 1.

CFBA was determined to have a Blaine fineness of 7220 cm^2^/g and a specific gravity of 3.05. Its main crystal phase was inferred to consist of quartz (SiO_2_), lime (free CaO), anhydrite (CaSO_4_), and calcite (CaCO_3_). Furthermore, a large number of hematite (Fe_2_O_3_) peaks have been detected in it. By comparison, atypical CFBBA has a crystal phase similar to that of CFBA and consists of anhydrite (CaSO_4_), quartz (SiO_2_), lime (free CaO), and Fe_2_O_3_. CFBBA has a higher free CaO content than that of CFBA, whereas its SO_3_ content is high because of desulfurization gypsum.

On the other hand, for typical pulverized fuel ash (PFA from Hadong Power of Korea Southern Power Co., Ltd., Busan, Republic of Korea), the SiO_2_ + Al_2_O_3_ + Fe_2_O_3_ content is higher than 80%, whereas the CaO content is less than 10%, which corresponds to Class F in the ASTM C618 standard. By comparison, in the case of the CFBC by-products CFBA and CFBBA, the SiO_2_ + Al_2_O_3_ + Fe_2_O_3_ content of each is 60.88% and 40.55%, respectively, whereas the CaO content is more than 5.5 times higher than the value dictated by the standard.

### 2.2. Method

Table 2 presents the experimental formulations of paste and mortar.

The heat of hydration was measured using a six-point multi-purpose conduction calorimeter from Tokyo-Rico Co., Ltd. in Tokyo, Japan, with the water/binder (W/B) ratio set to 0.5. Immediately after the addition of water, the heat of hydration was measured every 1 s for 60 s. The mixture was then stirred for 5 min, after which the heat of hydration was measured every 10 s for 24 h, and every 1 min for up to 72 h.

To examine the hardening properties, prismatic specimens (40 mm × 40 mm × 160 mm) with a water/cement (W/C) ratio of 0.62 were fabricated and subjected to water curing in accordance with the “compressive strength test method for cement mortar” described in KS L ISO 679 of the Korean Standards Association. The compressive strengths of the samples were then measured at 3, 7, and 28 days of age.

For the structural characteristics of the generated hydrates, the porosity was measured based on water tightness through mercury intrusion porosimetry (MIP) at a maximum pressure of 60,000 psi, and diameter range of 360 μm to 3 nm, using Auto Pore IV 9520 from Micromeritics in the United States.

Instrumental analysis was conducted to analyze the mechanisms of self-hardening and investigate the calcium carboferrite reactions for samples of varying ages (3, 7, 28, 56, 91, and 365 days). X-ray diffraction (XRD; D/Max-2500V from Rigaku, Tokyo, Japan) analysis was conducted at a scan rate of 4°/min in a 2θ range of 5 to 80°, whereas thermogravimetry/differential thermal analysis (TG-DTA; STA409PC Luxx model from Netzsch, Selb, Germany) was performed at a heating rate of 10 °C/min, up to 1000 °C, in a nitrogen (N2) atmosphere. In addition, field-emission scanning electron microscopy/energy-dispersive X-ray spectrometry (FE-SEM/EDS; JSM-6710F/X-Max model from JEOL/Tokyo, Japan) analyses were conducted on the elemental components, including C, O, Al, Si, S, Ca, and Fe, using 1000×, 5000×, and 10,000× magnifications, and EDS at an acceleration voltage of 15 kV and a working distance (WD) of 10 mm.

The specimens for analysis were fabricated from a paste with a W/C ratio of 0.5. A number of the specimens were subjected to air-dry curing (20 ± 1 °C, 60 ± 3% RH) for different durations and immersed in acetone. After the hydration process was stopped, the specimens were dried in a dryer (40 °C, 24 h), pulverized using a ball mill, sieved using sizes of 200 μm or less, and subjected to further analysis.

To analyze the mechanisms involving calcium carboferrite, an accelerated carbonation test was conducted on CFBA and CFBBA. A number of specimens were subjected to water curing at a temperature of 20 ± 2 °C for up to 28 days, after which the specimens were subjected to curing in a chamber at a constant humidity of 60 ± 5% and a constant temperature of 20 ± 2 °C until they reached 8 weeks of age. The specimens were then coated with epoxy resin on the pouring surface, bottom surface, and both sides, to block CO_2_. One of two carbonation methods was then applied. In the first method, the specimens were subjected to carbonation for either 28 or 56 days in a vacuum desiccator at a humidity of 60 ± 5%, temperature of 20 ± 2 °C, and CO_2_ gas concentration of 100%. In the second method, the specimens were exposed to the atmosphere, and measurements were performed after 28, 56, and 365 days in accordance with KS F 2596, “concrete carbonation depth measurement method”. The objective of this procedure was to examine the carbonation reaction due to the adsorption of CO_2_ from the atmosphere. The specimens were cut into 1 cm pieces, after which 1% phenolphthalein solution was sprayed onto the cut surfaces. Measurements were then performed from the surface to the reddish-purple section.

## 3. Result and Discussion

### 3.1. Strength Development Characteristics

CFBA and CFBBA were observed to have self-hydration and self-hardening properties. Figure 2 shows the compressive strengths of OPC, CFBA, and CFBBA according to age (3, 7, and 28 days). At 7 days of age, the CFBA and CFBBA mortars exhibited compressive strengths of approximately 10 and 8 MPa, respectively, which are approximately 30% of the compressive strength of the OPC mortar. At 28 days of age, they demonstrated compressive strengths of approximately 11 and 15 MPa, respectively.

In the case of the CFBA mortar, the compressive strength increased until 7 days of age, after which the rate of increase between 7 and 28 days of age became very low. This appears to be due to the higher fineness of CFBA compared to that of CFBBA. When the generated hydration products were analyzed, it was determined that ettringite and calcium aluminate hydrates (C-A-H) were generated by reactions among SO_3_, CaO, and Al_2_O_3_. Equation (1) shows the chemical reaction of CFBA and CFBBA. These hydrates were confirmed to have resulted in additional compressive strength, as in a previous study [19]. In the case of the CFBBA mortar, the compressive strength was lower than that of the CFBA mortar at early ages, but rapidly increased after 7 days of age. The compressive strength development due to the generation of ettringite was low at the early ages, apparently because the fineness of CFBBA was lower than that of CFBA. Afterward, the compressive strength developed more rapidly at the later ages because of the active generation of C-A-H due to hydration. Therefore, it was inferred that the compressive strength development with respect to time is influenced by the fineness of the material.
(1)$$\begin{array}{l} \;\mathrm{CaO}+H_{2}O\to {\left[\mathrm{Ca}{\left(\mathrm{OH}\right)}_{2}\cdot {\mathrm{Ca}}^{2+}\right]}^{20+}\cdot \left(n-x\right){\mathrm{OH}}^{-}\cdot {\mathrm{xOH}}^{-}\\ \mathrm{Ca}{\left(\mathrm{OH}\right)}_{2}+3{\mathrm{CaSO}}_{4}\cdot 2H_{2}O+{\mathrm{Al}}_{2}O_{3}+23H_{2}O\;\;\;\;\left(\mathrm{AFt}=\mathrm{ettringite}\right)\\ {\mathrm{SiO}}_{2}+\mathrm{xCa}{\left(\mathrm{OH}\right)}_{2}+\left(y-x\right)H_{2}O\to C_{x}{\mathrm{SH}}_{y}\;\;\;\;\left(\mathrm{CSH}\right) \end{array}$$

Table 3 shows the porosities and average pore diameters at 3 and 28 days of age. For CFBA, the pore distribution at 28 days was lower than that at 3 days. This finding, that the pore size and pore distribution decreased as the age increased, indicates that the pore structure was densified because of the generation of fine crystals. According to Metha and Monteiro, the pores can be classified based on size into gel micropores (<4.5 nm), mesopores (4.5 to 50 nm), middle capillary pores (50 to 100 nm), and large capillary pores (>100 nm) [20,21]. Figure 3 shows changes in pore size distribution between 3 and 28 days for CFBA and CFBBA. For CFBA, the number of large capillary pores decreased by approximately 66%, whereas the numbers of middle capillary pores, mesopores, and gel micropores increased by 9.17, 9.17, and 75.3%, respectively. By comparison, the porosity of CFBBA decreased by approximately 17.49%; specifically, the porosity was 46.22% at 3 days, and 38.14% at 28 days. This is due to the hydration products generated by the self-hardening property of CFBBA.

The compressive strength development of CFBA and CFBBA can be attributed to the densification of the structure inside the hardened body due to the generation of ettringite and C-A-H, the growth of ettringite and calcium aluminate crystals, and the generation of CaCO_3_ crystals. For OPC, C-S-H is the main hydrate that causes the compressive strength development. By contrast, in the case of CFBA and CFBBA, the main hydrate is C-A-H, which comprises single-phase crystals, unlike C-S-H, which comprises needle-like crystals. This appears to have been the cause of the relatively low compressive strength development in CFBA and CFBBA.

### 3.2. Heat of Hydration

In accordance with the experimental plan, Figure 4 compares the heats of hydration and cumulative heats of hydration of CFBA and CFBBA with those of OPC. It is known that the initial hydration of OPC is caused by the heat of the formation of ettringite generated by the reaction between gypsum and highly active aluminate and the heat of dissolution of the alite surface [22]. By contrast, in the case of CFBA, high heat of hydration occurs because of the generation of ettringite at early ages due to the aluminate and SO_3_ content. Among the clinker minerals of OPC, C_4_AF generates AFt (ettringite) and is converted into Afm (monosulfate) when gypsum undergoes a reaction. In the case of CFBA, the material contains a large amount of aluminate and approximately 11% Fe_2_O_3_, and thus the heat of formation increases with the reaction time, apparently because of additional reactions due to the presence of gypsum generated in the desulfurization process.

### 3.3. Characteristics Derived via Thermal Analysis

Figure 5 shows the thermal analysis results for the samples that have undergone hydration, with respect to the curing method (air-dry curing and steam curing) and age (3, 7, 28, 56, 91, and 365 days). From the thermal analysis results for CFBA, similar tendencies were observed for the samples between 3 and 91 days of age, whereas a different pattern was observed for the samples aged 365 days.

The hydration of CFBA was apparently at least 90% completed by 28 days of age. The samples that were at most 91 days of age exhibited rapid weight losses due to thermal decomposition between 100 and 200 °C. By contrast, the thermal analysis results for the samples aged 365 days show that weight loss gradually occurred up to approximately 700 °C. This indicates that the hydration of CFBA was completed after 91 days and that the material had been converted into a stable compound. For the samples aged 365 days, the weight losses between 450–550 and 750 °C were significant due to the curing of the material. It appears that curing resulted in large amounts of compounds that decompose at 750 °C or higher.

On the other hand, based on the thermal analysis results for CFBBA, the samples aged 3 and 7 days exhibited thermal decomposition of non-hydrated components, which occurs at early ages. Furthermore, the weight losses due to thermal decomposition were larger than those for CFBA. This seems to be due to the higher moisture (H_2_O) content of CFBBA compared to that of CFBA, which, in turn, is due to the larger particle size and higher porosity. The samples that were at most 91 days of age also exhibited rapid weight losses due to thermal decomposition between 100 and 200 °C. By contrast, the thermal analysis results for the samples aged 365 days show that weight loss gradually occurred up to approximately 700 °C. In addition, there were no weight losses after 800 °C, and the degree of weight loss for the samples aged 365 days was smaller than those demonstrated by the samples that were at most 91 days of age, which exhibited thermal decomposition of hydrates. For the samples aged 365 days, the weight loss between 450–550 and 750 °C was large. It appears that curing generated large amounts of compounds that decompose at 750 °C or higher.

### 3.4. Characteristics of Generated Hydrates

The crystal phases of the hydrates at 3, 7, 28, 56, 91, and 365 days of age are shown in Figure 6, whereas the image analysis results are shown in Figure 7 and Figure 8.

Ettringite, C-S-H, C-A-H, and calcite can be identified as the main hydration products of CFBA and CFBBA. In Figure 6, similar crystalline structures were observed for the samples that were at most 91 days of age. For the samples aged 365 days, it was inferred that the phase transition of ettringite occurred because of the consumption of gypsum.

Figure 7 shows the image analysis results. For the CFBA samples aged 3 days, the results were highly consistent with those of the XRD analysis. Highly active generation of ettringite and the presence of CaCO_3_ crystals and C-A-H were observed. By contrast, for CFBBA, the presence of gypsum and calcite, rather than that of ettringite, was confirmed. This tendency was exhibited by the samples that were at most 28 days.

On the other hand, at 365 days of age (Figure 8), CFBA exhibited the presence of CaCO_3_ crystals (circles) and C-A-H compounds (rectangles). Ettringite appears to have disappeared because of its transition to monosulfate. The transition of ettringite into monosulfate due to the consumption of gypsum is the general theory for the hydration process. The FE-SEM/EDS analysis results indicate the presence of sulfur ions. The presence of a monosulfate peak was not easily confirmed because of overlapping peaks even though a transition to monosulfate had occurred. By contrast, at 365 days of age, CFBBA exhibited the presence of aragonite crystals (red), which are needle-shaped calcium carbonates; calcite crystals (blue), which are hexagonal CaCO_3_ crystals; and monosulfate crystals (yellow).

### 3.5. Accelerated Carbonation

To investigate the characteristics of calcium carbonation reactions for CFBA and CFBBA, a test was conducted in accordance with KS F 2584, “accelerated carbonation test method for concrete”.

The accelerated carbonation test results for steam-cured OPC show that carbonation occurred at the fourth week of the accelerated carbonation process, and that carbonation occurred at the 56th day of the carbonation process regardless of the curing method. In the case of the specimens left outdoors unattended, carbonation did not occur even for a year. It seems that the outdoor CO_2_ concentration was exceedingly low for OPC to undergo changes. By contrast, for all CFBA and CFBBA specimens, carbonation occurred during the accelerated carbonation process regardless of the curing method. For CFBA, carbonation occurred on all specimens exposed to the outside, whereas for CFBBA, carbonation occurred on more than 90% of the specimens.

As in the thermal analysis results presented earlier, the increase in weight loss due to the decomposition of calcite with respect to increasing age indicates the generation of carbonate. The specimens left outdoors unattended exhibited the largest difference between CFBA and CFBBA. For CFBA, all the specimens were subjected to carbonation as the age increased regardless of the curing method and curing period. In the case of CFBBA, all of the specimens subjected to the accelerated carbonation process underwent carbonation, whereas slow carbonation occurred on the specimens exposed to the outside, as shown in Table 4.

The results of this study show that both CFBA and CFBBA exhibit CO_2_ adsorption, which can contribute to compressive strength development via the generation of CaCO_3_, which densifies the pores of the hardened body.

## 4. Conclusions

Circulating fluidized-bed boiler fly ash (CFBA) and circulating fluidized-bed boiler bottom ash (CFBBA), which are by-products of the circulating fluidized-bed boiler combustion (CFBC) of coal, contain Fe_2_O_3_, free CaO, and CaSO_4_ in large quantities. In this study, the compressive strength development due to the generation of ettringite (Ca_6_Al_2_(SO_4_)_3_(OH)_2_·26H_2_O) and carbo-ferrite hydrates (3CaO·Fe_2_O_3_·CaCO_3_·12H_2_O) or carbo-aluminate hydrates (3CaO·Al_2_O_3_·CaCO_3_·12H_2_O) during the hydration reaction of CFBA and CFBBA was analyzed. In addition, the self-hydration and carbonation reaction mechanisms of CFBC were investigated. The following findings were derived from this study.

(1)For CFBA and CFBBA, reactions among SO_3_, CaO, and Al_2_O_3_ generate ettringite and calcium aluminate hydrates (C-A-H), thereby developing the compressive strength of the material. When the crystal phases of hydrates were analyzed, CFBA exhibited very active generation of ettringite and the presence of CaCO_3_ crystals and C-A-H at 3 days of age. By contrast, for CFBBA, the presence of gypsum and calcite, instead of that of ettringite, was confirmed.(2)At 365 days of age, CFBA exhibited the presence of CaCO_3_ crystals and C-A-H compounds, whereas CFBBA exhibited the presence of C-A-H compounds; aragonite crystals, which are needle-shaped calcium carbonates; calcite crystals, which are hexagonal CaCO_3_ crystals; and monosulfate crystals. The reduction in pores inside the hardened body densifies the pore structure through the generation of hydrates, such as ettringite, C-A-H, carbo-ferrites, and CaCO_3_.(3)The results of the accelerated carbonation test show that both CFBA and CFBBA were subjected to carbonation regardless of the curing method. In the carbonation procedure through outside exposure, all the CFBA specimens underwent carbonation, whereas more than 90% of the CFBBA specimens underwent carbonation.

In conclusion, CFBA and CFBBA attain hardening properties because of the hydration reaction of calcium aluminate (and/or ferrite) due to calcium aluminate ferrite and sulfate, and the carbonation reaction caused by CO_2_ absorption of calcium aluminate ferrite. This can be identified as a self-hardening process because it does not require a supply of special ions from the outside. It was confirmed that CFBA and CFBBA improve their compressive strengths via the densification of the pores of the hardened body through the formation of CaCO_3_ by carbonation of CO_2_ absorption.

## Figures and Tables

**Figure 1 materials-16-05498-f001:**
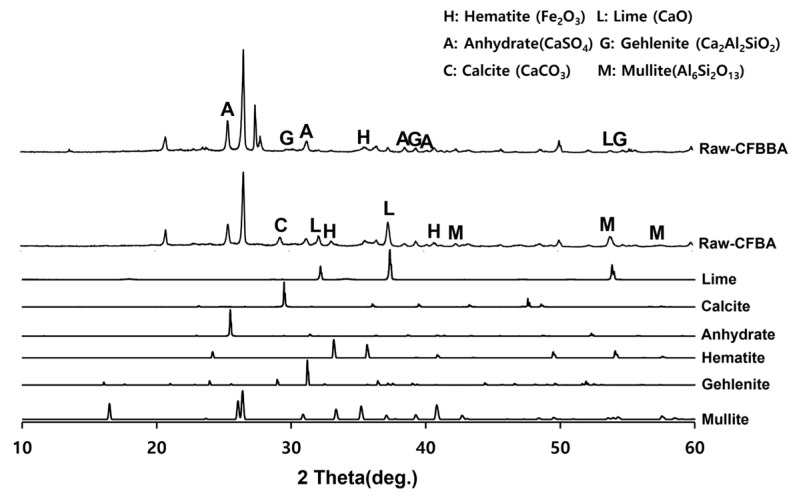
XRD pattern of CFBA and CFBBA.

**Figure 2 materials-16-05498-f002:**
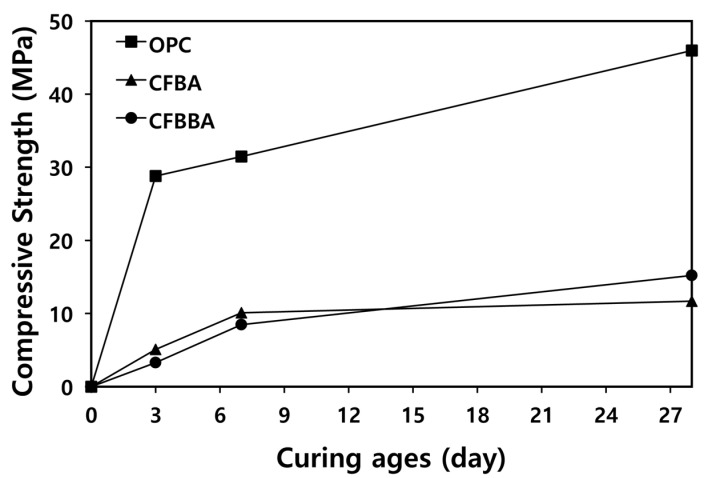
Compressive strength of mortar with OPC and CFBC by-productions.

**Figure 3 materials-16-05498-f003:**
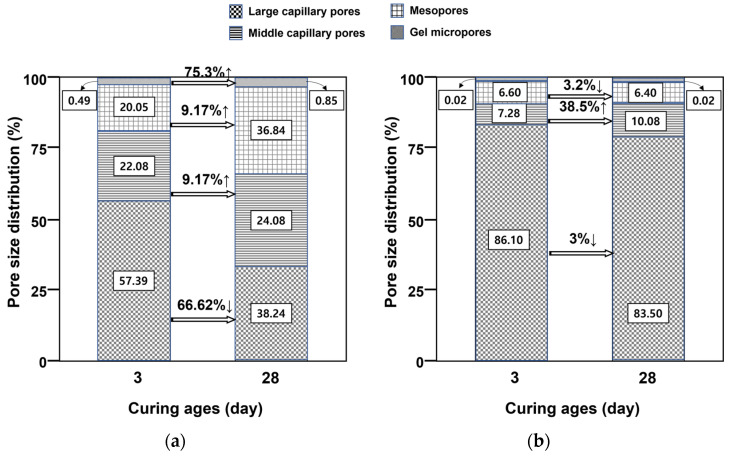
Variation in distribution by pore size with curing age: (**a**) CFBA. (**b**) CFBBA.

**Figure 4 materials-16-05498-f004:**
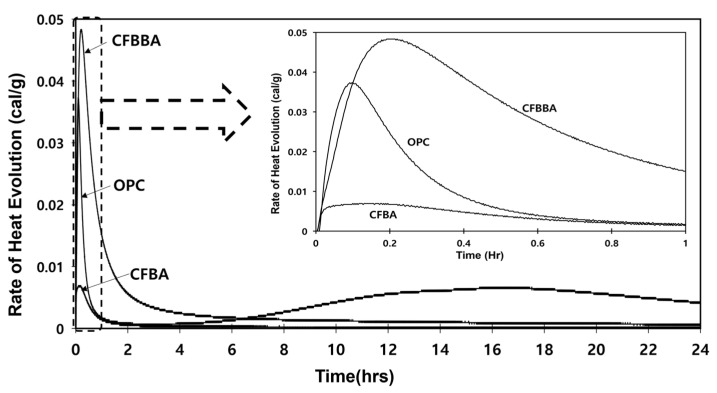
Isothermal conduction calorimeter of sample.

**Figure 5 materials-16-05498-f005:**
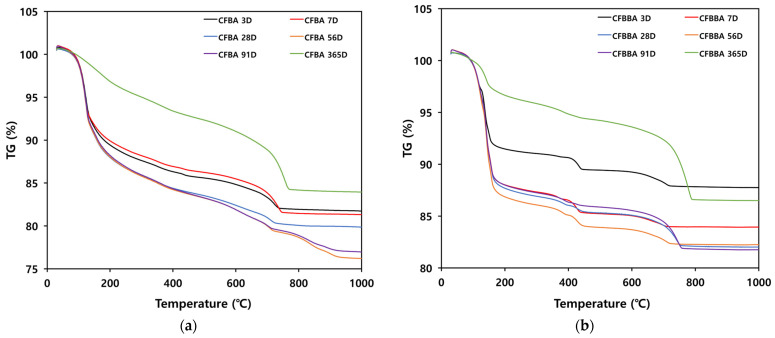
TG curves of CFBA and CFBBA with curing time: (**a**) CFBA. (**b**) CFBBA.

**Figure 6 materials-16-05498-f006:**
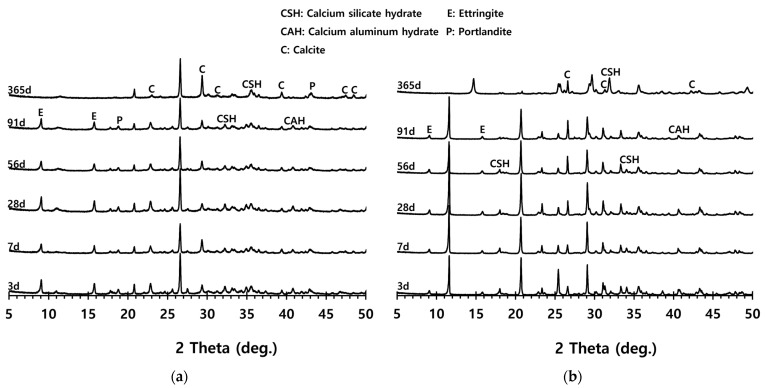
XRD pattern of CFBA and CFBBA with curing ages: (**a**) CFBA. (**b**) CFBBA.

**Figure 7 materials-16-05498-f007:**
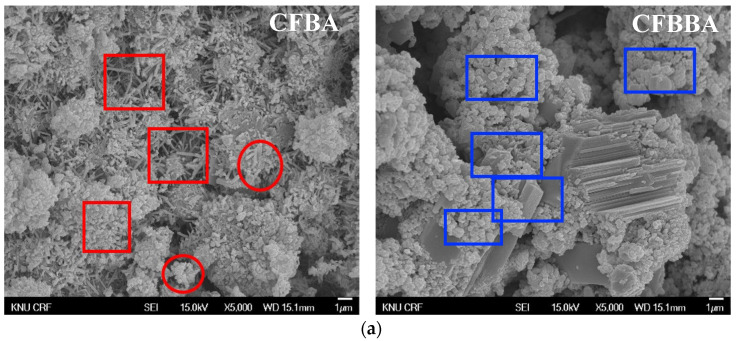
SEM of CFBA and CFBBA at 3 days and 28 days: (**a**) 3 days. (**b**) 28 days.

**Figure 8 materials-16-05498-f008:**
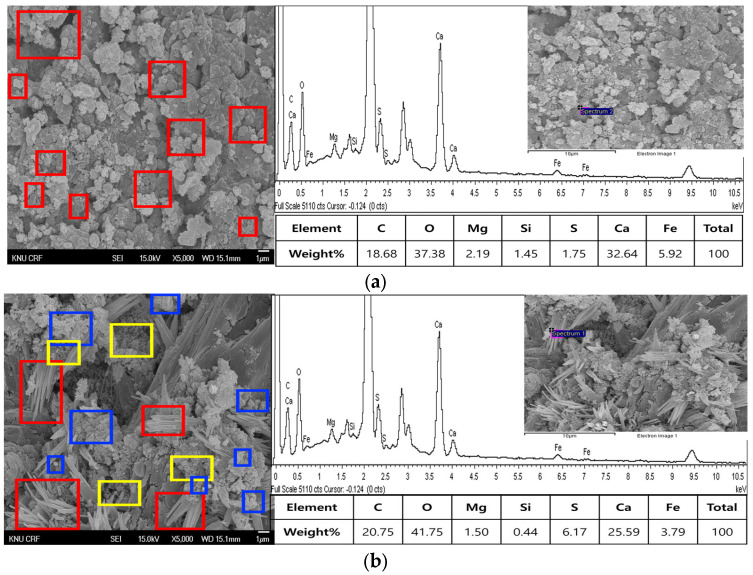
FE-SEM/EDS analysis of CFBA and CFBBA hydration at 365 days: (**a**) CFBA. (**b**) CFBBA.

**Table 1 materials-16-05498-t001:** Chemical composition and physical property of CFBA, CFBBA, and PFA.

Type	Component	Materials		
		PFA	CFBA	CFBBA
Chemical (%)	CaO	4.15	22.69	27.71
	SiO_2_	60.25	33.33	25.26
	Al_2_O_3_	22.15	16.64	7.44
	Fe_2_O_3_	4.0	10.91	7.85
	MgO	1.59	3.86	3.92
	K_2_O	1.48	1.64	0.94
	SO_3_	1.10	8.55	24.27
	Free-CaO		3.57	1.65
	Loss on ignition	4.55	2.38	2.61
Physical	Blaine (cm^2^/g)	3570	7220	838
	Specific gravity	2.29	3.05	2.95

**Table 2 materials-16-05498-t002:** Mixing ratio.

	Paste			Mortar		
	1	2	3	1	2	3
OPC	1	0	0	1	0	0
CFBA	0	1	0	0	1	0
CFBBA	0	0	1	0	0	1
Sand	0	0	0	3	3	3
W/C	0.5	0.5	0.5	0.62	0.62	0.62

**Table 3 materials-16-05498-t003:** Porosity and average pore diameter at Curing ages (3, 28 days).

Type	Porosity (%)		Average Pore Diameter (nm)	
	3 Day	28 Day	3 Day	28 Day
CFBA	59.2	59.2	55.2	38.5
CFBBA	46.2	38.1	191.4	177.6

**Table 4 materials-16-05498-t004:** Carbonation of OPC, CFBA, and CFBBA.

Type	Curing Method	Curing Age					
		28 d		56 d		365 d	
		Air	Steam	Air	Steam	Air	Steam
OPC	A.C ^(1)^	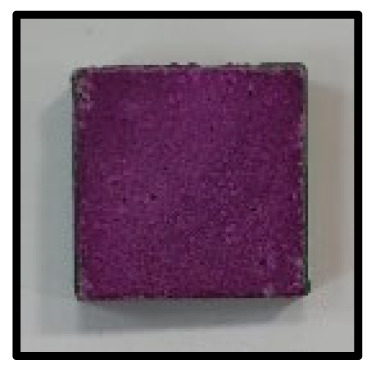	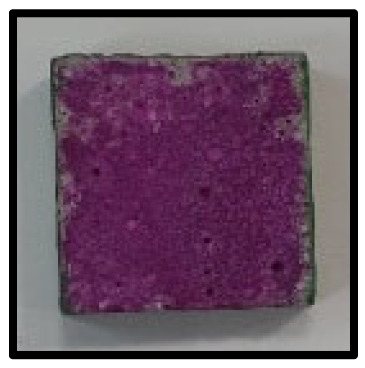	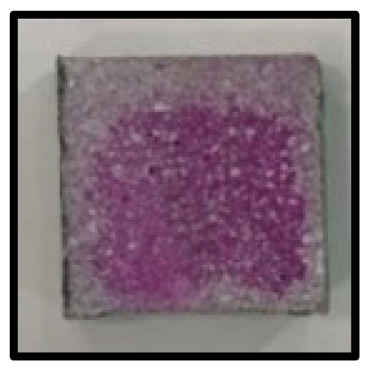	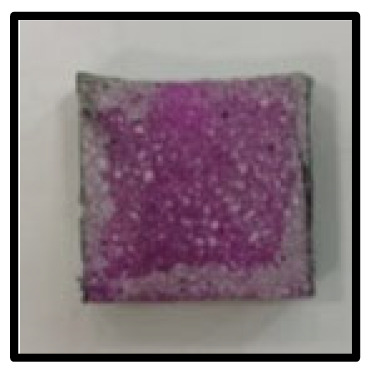		
	C.A.P. ^(3)^ (%)	0	10.8	39.5	34.1		
	O.S ^(2)^	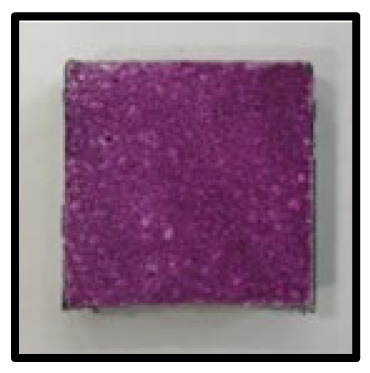	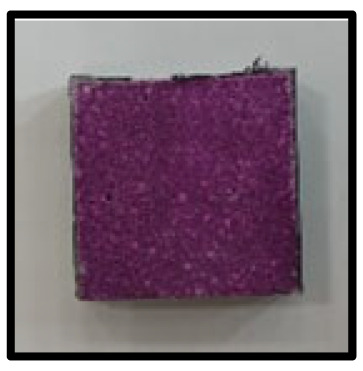	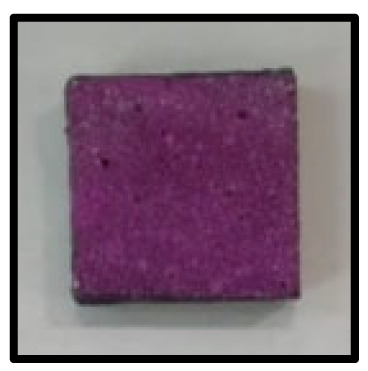	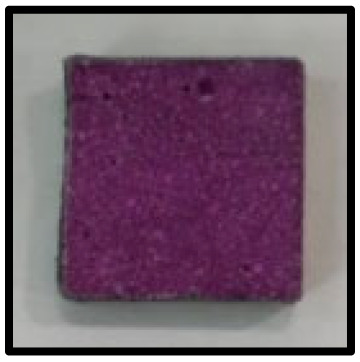	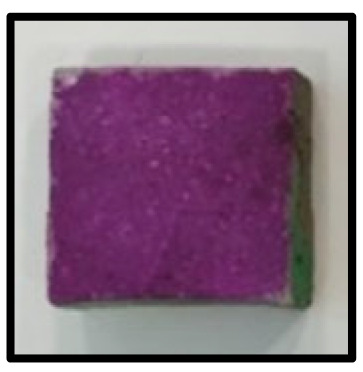	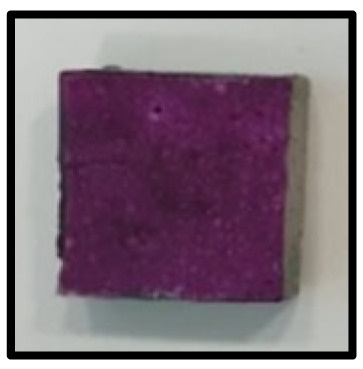
	C.A.P. (%)	0	0	0	0	0	0
CFBA	A.C	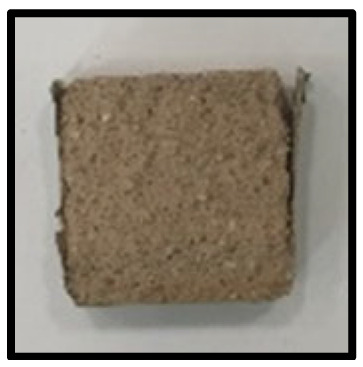	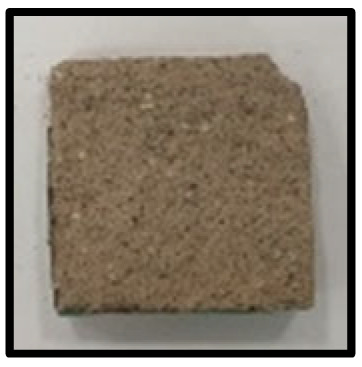	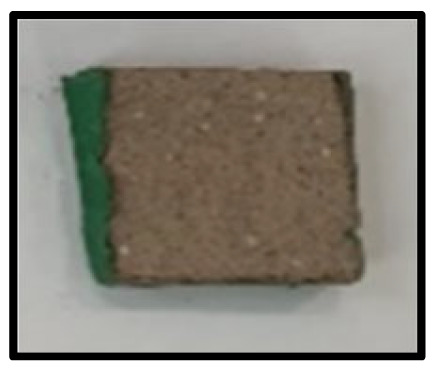	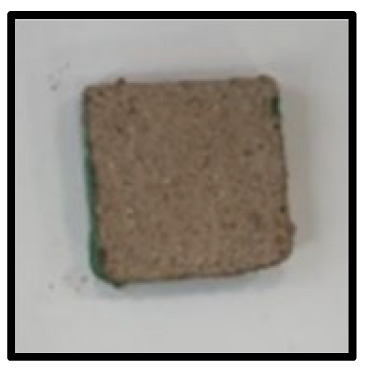		
	C.A.P. (%)	100	100	100	100		
	O.S	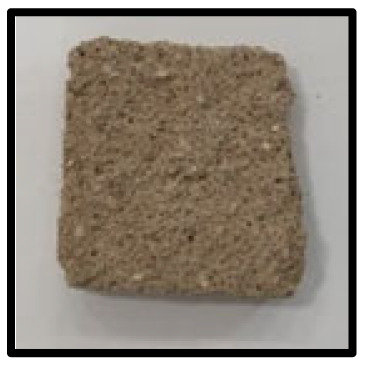	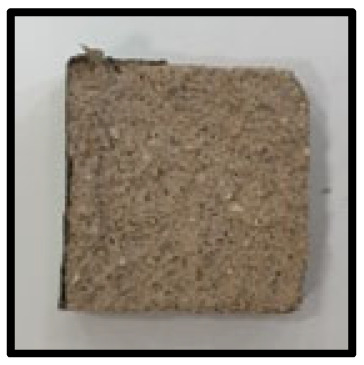	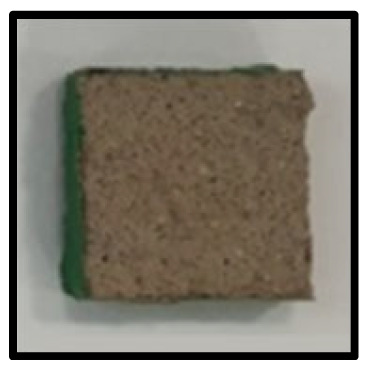	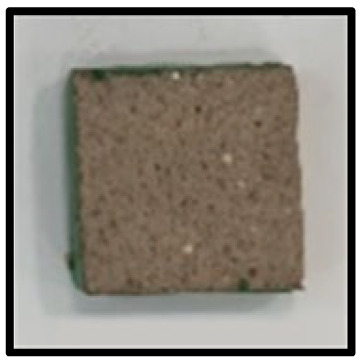	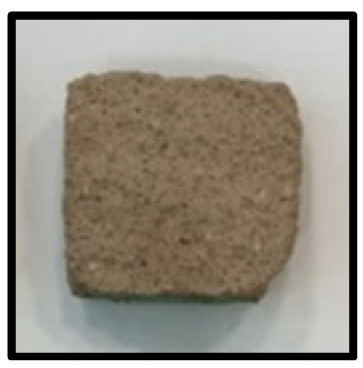	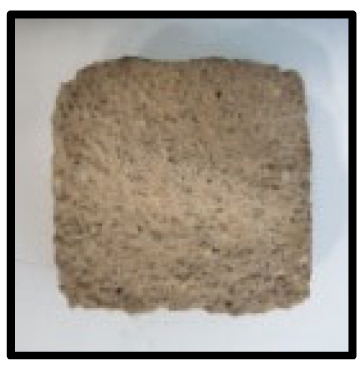
	C.A.P. (%)	100	100	100	100	100	100
CFBBA	A.C	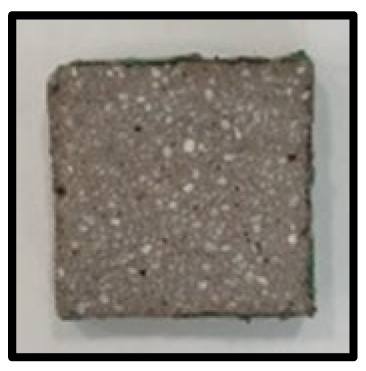	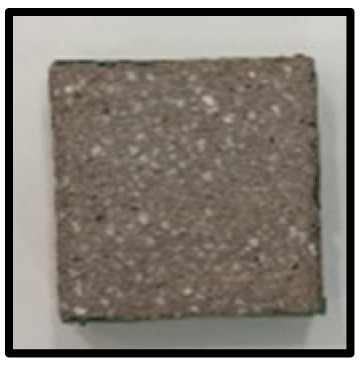	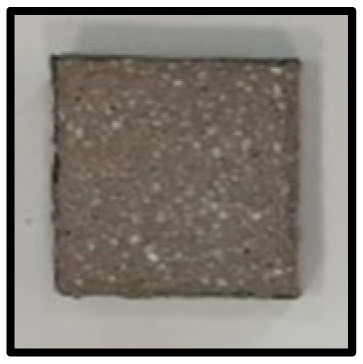	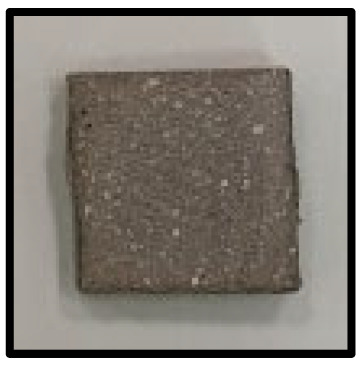		
	C.A.P. (%)	100	100	100	100		
	O.S	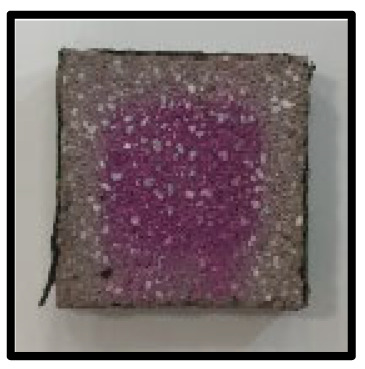	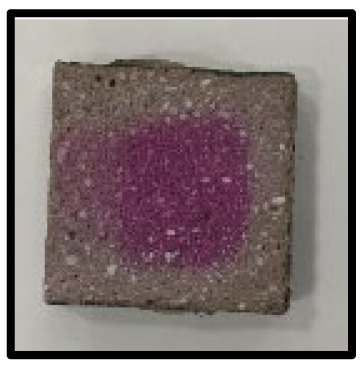	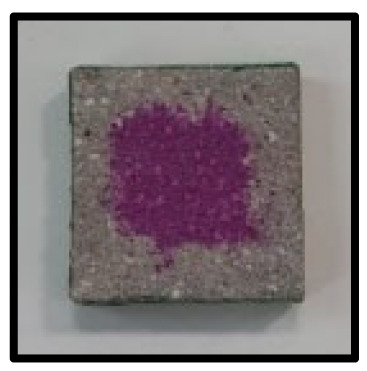	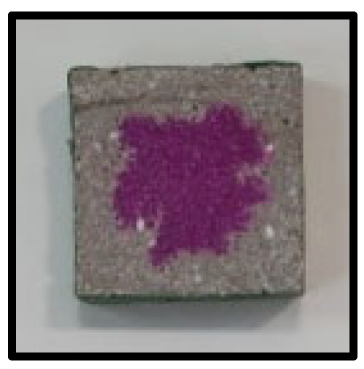	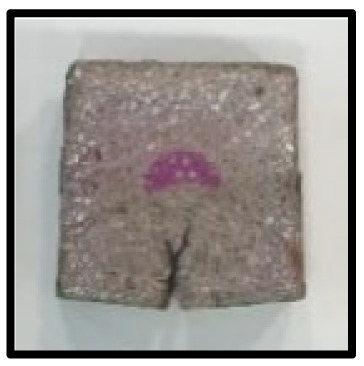	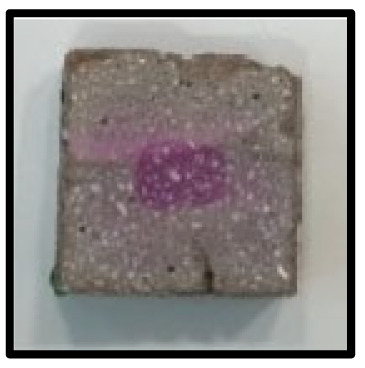
	C.A.P. (%)	55.0	58.1	65.5	64.6	95.9	90.7

^(1)^ AC: Accelerated carbonation. ^(2)^ OS: Outside. ^(3)^ Carbonation area percent.

## Data Availability

Not applicable.

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
