# Peer review of "The Characteristics of Self-Hydration and Carbonation Reaction of Coal Ash from Circulating Fluidized-Bed Boiler by Absorption of CO2"

_materials, 2023, doi:10.3390/ma16155498_

Round 1

Reviewer 1 Report

The general impression is that the present version of the paper is made offhand. Nevertheless, I am suggesting that authors can try to improve the Ms.

Lines 32-33 It is not clear if there are limitation for use CFBC products why was the present investigation made. What kind of new information was obtained? How the materials can be used?

51-51 Space before formula

67 Provide the source of fuel ash.

71 How were the values of oxides ratios obtained? They are different from ones shown in the Table 1.

Table 1. What is F-CaO? Is the blaine value of 838 correct?

Figure 1 and all other figures. Indicate the figures related to CFBC and CFBBC samples. Gypsum is the major phase of CFBBC but it does not indicated (but exists in caption).

149-151 Arrange correctly superscripts. The line of Aft is completely unclear. And only much later the meaning of Aft is shown. What is the meaning of oxides summa?

 173. It seems to me it has to be the figure 2.

Figure 3. I did not found gel micropores. Line 176, picture b – Correct the line under the figure.

3.2. There is nothing about hydration and high heat of hydration of CFBBC sample. Why it so high? Why it so much differs from CFBC sample?

Figure 4. I am suggesting time (hrs).

3.3. Here are very similar data on both samples. It is better to use the same y-axis for both figures 5a,b. The question is rising again about anomalous heat of hydration of the sample CFBBa.

Figure 5. What is the meaning of A before the cod of the samples?

Line 235. Not clear

Figure 6b. CAH is shown in caption but not indicated in the figure.

Figure 7. What is the meaning of boxes and ovals? Indicate minerals. In the caption SEM data are mentioned. Where are such data?

248 In the text is the reference on CFBBA sample, but in the caption the reference on CFBA sample. What is correct?

Figure 8. Check the meaning of boxes and ovals and better indicate mineral phases on the photos. Indicated on SEM spectra the possible mineral phases. 

Table 5. What is the meaning of A.C, C.A.P. and O.S. ? More details describing these table are needed.

284 and 289 Subscripts

Conclusion. What kind of new information was obtained? How the studied materials can be used?

Author Response

Thank you so much for spending your time for my thesis. Thanks to you, I have been able to rethink my thesis and improve the quality of my thesis. I am not good at English, so please understand if I am not polite in answering. Thank you again.

Reviewer 2 Report

Lee et al. have put together a Research article on the self-hydration and carbonation reaction of coal ash promoted by CO2 adsorption. The phase transition is characterized properly via TGA, SEM, etc.. The reviewer recommends the article to be published in Materials with minor revision.

Some minor points need to be addressed:

1. Please add the XRD patterns of standard materials (H, G, A, L, C) as references to the bottom of Figure 1.

2. Table 3 is unnecessary to be included as details are discussed in the Method section.

3. The reviewer finds difficult to tell the differences of each trace in Figure 5. Please use color code to discriminate those traces.

4. Please highlight the phase evolution in Figure 6 with circle.

5. Figures 7 and 8 are very crowded with boxes. Each species only needs 1 box.

Author Response

(The authors gave the same response as above.)

Round 2

Reviewer 1 Report

Several new comments:

Lines 60-65 What phase is related to Al2O3? Show it on the figure 1.

Line 71 How is 60.88 half of 80?

Table 1. Are you definite that blaine fineness of CFBBA 9 times below than one of CFBA?

Author Response

Thank you for your comment.

I have finished editing what you checked.

  1. I plotted against aluminum oxide in the graph.
  2. A correction was made to the content.
  3. Table 1, That's right. Unlike general bottom ash, it has thin sand-like particles.

Thank you for your comments and any other comments are welcome.